# Changes in the Ultrastructure of *Staphylococcus aureus* Treated with Cationic Peptides and Chlorhexidine

**DOI:** 10.3390/microorganisms8121991

**Published:** 2020-12-14

**Authors:** Alina Grigor’eva, Alevtina Bardasheva, Anastasiya Tupitsyna, Nariman Amirkhanov, Nina Tikunova, Dmitrii Pyshnyi, Elena Ryabchikova

**Affiliations:** Institute of Chemical Biology and Fundamental Medicine, Siberian Branch of Russian Academy of Science, Lavrent’ev av. 8, 630090 Novosibirsk, Russia; feabelit@mail.ru (A.G.); herba12@mail.ru (A.B.); aysa@ngs.ru (A.T.); nariman@niboch.nsc.ru (N.A.); tikunova@niboch.nsc.ru (N.T.); pyshnyi@niboch.nsc.ru (D.P.)

**Keywords:** *S. aureus*, transmission electron microscopy, cationic peptides, chlorhexidine, cell membrane, cell wall, cytoplasm damage

## Abstract

Antimicrobial peptides, including synthetic ones, are becoming increasingly important as a promising tool to fight multidrug-resistant bacteria. We examined the effect of cationic peptides H_2_N-Arg_9_-Phe_2_-C(O)NH_2_ and H_2_N-(Lys-Phe-Phe)_3_-Lys-C(O)NH_2_ on *Staphylococcus aureus*, which remains one of the most harmful pathogens. Antiseptic chlorhexidine served as reference preparation. We studied viability of *S. aureus* and examined its ultrastructure under treatment with 100 µM of R9F2 or (KFF)3K peptides or chlorhexidine using transmission electron microscopy of ultrathin sections. Bacterial cells were sampled as kinetic series starting from 1 min up to 4 h of treatment with preparations. Both peptides caused clearly visible damage of bacteria cell membrane within 1 min. Incubation of *S. aureus* with R9F2 or (KFF)3K peptides led to cell wall thinning, loss of cytoplasm structure, formation of mesosome-derived multimembrane structures and “decorated fibers” derived from DNA chains. The effect of R9F2 peptides on *S. aureus* was more severe than the effect of (KFF)3K peptides. Chlorhexidine heavily damaged the bacteria cell wall, in particular in areas of septa formation, while cytoplasm kept its structure within the observation time. Our study showed that cell membrane damage is critical for *S. aureus* viability; however, we believe that cell wall disorders should also be taken into account when analyzing the effects of the mechanisms of action of antimicrobial peptides (AMPs).

## 1. Introduction

*Staphylococcus aureus (S. aureus),* which causes a wide variety of pathologies, remains one of the most harmful bacteria, and has been researched for over 100 years. The notorious “reputation” of *S. aureus* is primarily based on its resistance to various groups of conventional antibiotics [1,2,3,4], and the need to obtain a new drugs which effectively destroy *S. aureus* and do not cause the development of drug resistance is among the top priorities for effective control of the infection [5,6]. Antimicrobial peptides (AMPs) are considered as promising tool to fight multidrug-resistant bacteria. Currently, more than 5000 of AMPs are known; these small molecules are found in all organisms and operate as a part of the innate immune response, killing bacteria, viruses, fungi and even tumor cells [7,8,9]; natural AMPs effective against *S. aureus* were reviewed in [3]. Although many details of natural peptide action are still unknown, the collected data show that their effect is not associated with specific bacteria molecules, and therefore the development of bacterial resistance to AMPs may be a rare phenomenon, which increases the significance of these compounds [10]. The exceptional antimicrobial properties of natural AMPs are realized in living organisms, but their practical use in medicine is hampered by the difficulty of obtaining AMPs from natural sources and standardization for medical application, so synthetic AMPs are entering the arena as the “main players” [9,11].

A number of models for the interaction of bacteria with AMPs have been proposed, but the mechanism of this interaction is not fully understood [7,8,9]. Meanwhile, it is known that AMPs interact with bacteria cell membranes, and such physico-chemical properties as amino acid sequence, length, helicity, net charge, hydrophobicity, amphipathicity and solubility are important for AMPs’ effect on bacteria cells [7,12]. It is believed that synthetic peptides can be highly effective and affordable analogs of natural AMPs, and therefore understanding the mechanisms of their effects on bacterial cells is necessary to optimize their procurement and increase their efficiency.

Evaluation of the mechanisms of AMP action could be made at different levels: whole population of cells, single cells and at molecular level. The action of AMPs at the population level is usually studied using microbiological and biochemical methods. Transmission Electron Microscopy (TEM) represents the main tool for studying changes at the cellular level. This method allows to analyze cell structures at the nanometer scale and gives an idea of what is happening inside the cell [13,14]. In contrast, scanning electron microscopy (SEM) allows to examine the bacteria surface only and to see cell wall deformation or breach, and leakage of the cytoplasm [15,16,17].

Earlier we synthesized and characterized two cationic AMPs, highly amphiphilic R9F2 and moderately amphiphilic peptide (KFF)3K, and showed their antibacterial effect, as well as the absence of cytotoxicity for eukaryotic cells [18]. We applied TEM of ultrathin sections for examination of successive changes in *Candida albicans* ultrastructure under the influence of R9F2 and (KFF)3K peptides and found clear differences in their effects [19]. *C. albicans* is an eukaryotic microorganism with a thick cell wall, mainly composed of highly mannosylated glycoproteins [20], and it was interesting to compare its damage by R9F2 and (KFF)3K peptides with their effects on prokaryotic microorganisms enclosed in a cell wall, such as *S. aureus*. The search for publications providing TEM data on the details of AMPs interaction with *S. aureus* cells was unsuccessful: the works reported microbiological studies of peptide efficacy, presented AMP physico-chemical characteristics, discussed possible mechanisms of AMPs action, and sometimes showed damage of the cells in TEM or SEM at the last time-point of incubation [21,22,23,24,25].

To compare the changes in *S. aureus* ultrastructure caused by cationic peptides with some well-known antibacterial compounds (positive control), we decided to use chlorhexidine as a reference, confident that its effects were well studied and published. To our surprise, only 35 publications were found in PubMed for the query “chlorhexidine & electron microscopy & *S. aureus*”, and none of them provided information on the effect of chlorhexidine itself or on the ultrastructure of this bacterium. Thus, it turned out that in order to compare the effect of peptides with the effect of chlorhexidine, it was necessary to study in detail the effect of the latter on *S. aureus* ultrastructure.

We focused our work on how the R9F2 and (KFF)3K peptides affect *S. aureus* cells, using analysis of bacteria ultrastructure on ultrathin sections. We examined the chlorhexidine effect on *S. aureus* cells and described its ultrastructural characteristics. We also compared features of cell wall damage in *S. aureus* with the previously studied cell wall in *C. albicans* to find similarities and differences.

## 2. Materials and Methods

### 2.1. Peptides

The peptides H_2_N-(Lys-Phe-Phe)_3_-Lys-C(O)NH_2_ and H_2_N-Arg_9_-Phe_2_-C(O)NH_2_ were synthesized at >95% purity and verified by mass-spectrometry (Appendix A), in the text they are designated as peptides (KFF)3K and R9F2 correspondingly. Data on the physico-chemical properties of these peptides were published earlier [18]. The R9F2 peptide possesses pronounced amphiphilic properties due to maximal spacing of the cationic and hydrophobic parts along the peptide chain. The hydrophobic residues and positive charge in the (KFF)3K peptide molecule are uniformly alternating and distributed within the linear chain, so its linear cationic-hydrophobic polarity is almost aligned. The peptides (KFF)3K and R9F2 differ in net charge (+5 and +10), and the first, according to data of reverse-phase high-performance liquid chromatography (RP-HPLC), is more hydrophobic in aqueous media [18].

### 2.2. Microorganism and Growth Conditions

*S. aureus* ATCC 25923 strain was obtained from American Type Culture Collection and was stored at the Collection of Extremophile Microorganisms and Type Cultures of ICBFM SB RAS (Novosibirsk, Russia). The bacteria was stored at −70 °C in a NaCl-free Luria-Bertani BD (LB, Difco, Franklin Lakes, NJ, USA) broth with 25% glycerol. Before experiments, the culture was plated on LB agar and incubated at 37 °C for 16 h. Then, two µL of the culture were sown by bacteriological loop in 100 mL of LB broth and incubated for 16 h at 37 °C. After the incubation, bacteria cells were concentrated at 10,000 rpm for 10 min using Eppendorf 5810R centrifuge (Eppendorf, Vienna, Austria,), then supernatant was discarded and the cell concentration was adjusted to 1 × 10^6^ CFU/mL by adding fresh LB broth. The resulting cell suspension was added to a 96-well plate in the volume of 200 µL per well. The optical density was measured on a flatbed reader Uniplan (Picon, Moscow, Russia) at a wavelength of 595 nm; data are shown on Appendix A.

### 2.3. Effect of (KFF)3K and R9F2 Peptides on S. aureus Growth

*S. aureus* cells were incubated for 16 h in LB broth (beginning of stationary phase) and sedimented at 10,000 rpm for 10 min using Eppendorf 5810R centrifuge (Eppendorf, Vienna, Austria). Pellets were resuspended up to concentration 1 × 10^8^ CFU/mL in 0.9% of NaCl solution (saline) containing 100 or 200 µM of R9F2 or (KFF)3K peptides, the resulting suspensions were incubated at 37 °C. The same protocol was used for treatment of *S. aureus* with 100 or 200 µM of chlorhexidine, serving as a positive control. The cells incubated in saline or LB broth served as negative control (intact) cells.

To evaluate the effect of preparations, the samples (0.1 mL) were collected after 0, 15, 30, 45, 60, 120 and 240 min and were sown on LB agar to determine number of colonies after 16–18 h of incubation at 37 °C according to [26].

### 2.4. Processing of the Samples for TEM Studies

*S. aureus* cells were incubated for 16 h in LB broth and sedimented at 10,000 rpm for 10 min using Eppendorf 5810R centrifuge (Eppendorf, Vienna, Austria). Pellets were resuspended up to concentration 1 × 10^8^ PFU/mL in saline containing 100 µM of R9F2 or (KFF)3K peptides, or chlorhexidine, to a final volume of 10 mL, the resulting suspensions were incubated at 37 °C. The cells incubated in saline served as a negative control (designated as “control” or “normal” cells in the text). The samples were collected after 0, 15, 30, 45 min and 1, 2 and 4 h of incubation.

Suspensions were sedimented at 8000 rpm for 5 min, then supernatants were removed and loose pellets were suspended in 2 mL of 4% paraformaldehyde and 2.5% glutaraldehyde mixture (2:1). Subsequently suspensions were sedimented for 10 min at 10,000 rpm using Eppendorf 5810R centrifuge (Eppendorf, Vienna, Austria) and left for 24 h at 4 °C. After that, the samples were washed from fixative and postfixed with 1% osmium tetraoxide solution for 1 h, routinely dehydrated in ethanol and acetone, and embedded in an epon-araldite mixture to obtain hard blocks. All reagents for TEM processing were purchased from EMS (Hatfield, PA, USA).

Ultrathin sections were prepared on an ultramicrotome EM UC7 (Leica, Wetzlar, Germany) using a diamond knife (Diatome, Nidau, Switzerland), and contrasted with 2% water solutions of uranyl acetate and lead citrate. The sections were examined in a JEM 1400 TEM (JEOL, Tokyo, Japan). Digital images were collected using a Veleta side-mounted camera (EM SIS, Muenster, Germany).

Ultrathin sections are 70–80 nm in thickness, and can pass the cells in various directions, causing a visible polymorphism of *S. aureus* cells. We analyzed cells and their structures on cross sections only. The thickness of cell wall and cell membrane were measured only in areas sectioned perpendicularly to these structures.

## 3. Results

### 3.1. Viability of S. aureus Cells Incubated with R9F2 or (KFF)3K Peptides or Chlorhexidine

Previously, we determined the main antibacterial characteristics of R9F2 and (KFF)3K peptides for gram-positive microorganism *S. aureus*. The peptides showed dose-dependent activity against the bacterium, the values of minimal inhibitory concentration (MIC) were 1 ± 0.3 and 13.5 ± 4.1 µM for R9F2 and (KFF)3K peptides, respectively. Thus, the “killing” ability of peptides differed by more than ten times [18]. The present work focuses on how the cationic peptides R9F2 and (KFF)3K interact with *S. aureus* at the individual cell level using TEM of ultrathin sections. To provide the best visualization of the effect of peptides by this technique, we used relatively high peptide concentrations; saline instead of culture medium (to avoid interaction of peptides with medium components); and a four hour observation period.

Incubation of *S. aureus* cells in saline did not change the viability of the cells compared with LB broth within 24 h (Figure 1). Both peptides and chlorhexidine caused clear dose-dependent reduction of *S. aureus* viability within 4 h (Figure 1). The fast and most pronounced drop in bacteria viability was detected when *S. aureus* was incubated in the presence of 200 µM of R9F2 peptide. The effect of this peptide in concentration of 100 µM was also clearly expressed, and by values at different points it was close to the effect of (KFF)3K peptide in a concentration of 200 µM. The latter, in a concentration of 100 µM, caused a slight decrease in bacteria viability, similar to those caused by chlorhexidine in a concentration of 100 µM. Changes in the viability of *S. aureus* under the influence of chlorhexidine at a concentration of 200 µM showed a slow decrease during the first two hours and a sharp drop between 2 and 4 h of incubation, the viability value at the 4 h point was equal to the value shown by the peptide R9F2. The viability curves for the R9F2 peptide were straight and showed a proportional dependence on the incubation time, while the curves of the other drugs were broken (Figure 1).

### 3.2. Ultrastructure of S. aureus Cells Incubated in Saline (Control)

Ultrastructure of *S. aureus* cells incubated in saline did not noticeably differ from published data [15,27,28]. *S. aureus* cells in ultrathin sections (Figure 2A and Figure 4C) had a shape close to spherical and a diameter of 700–800 nm, about half of cells showed the formation of septa, indicating cell division.

Cell walls (Figure 2) were 20–25 nm in thickness and of middle electron density with dense outer edge; its relief gave idea on the bacteria surface, which can be smooth or rough. Electron density of the outer edge is associated with the presence of teichoic acids, which have a higher affinity for the contrasting agents than peptidoglycans composing a cell wall [29].

An intermediate layer (Figure 2 and Figure 3) of a high electron density and a thickness of 6–7 nm, was located between the inner edge of cell wall and the cell membrane, a space occupied by an intermediate layer is called “periplasmic space”, and the intermediate layer itself in some works is called “periplasm” [30]. Intermediate layer consisted of granular material; fine “grains” of 1–2 nm were chaotically located (Figure 2B and Figure 3A). The periplasmic space contains precursors of peptidoglycan, fragments of teichoic acid, membrane-bound proteins and ions, which makes the space critically important for the formation and functioning of the cell wall [29,31,32].

Cell membrane of *S. aureus* (Figure 2B) was recognized by a clearly visible middle lipid electron-transparent layer (~4 nm thick); the outer electron-dense layer merged with the above mentioned intermediate layer, and was indiscernible. The internal electron-dense layer of the cell membrane was also poorly distinguishable due to electron density equal to that of the cytoplasm. The cytoplasm (Figure 2) had an average electron density and contained ribosomes; some areas were devoid of ribosomes and had a fine-grained structure. Few control cells of *S. aureus* contained mesosomes, which were folds of the cell membrane (Figure 2D).

The area occupied by a nucleoid in a cell on ultrathin sections could be distinguished by the presence of an irregularly shaped electron-transparent “core” or structureless electron dense material surrounding this “core” like “a shell” (up to 60 nm in thickness). Poorly distinguishable DNA strands sometimes were seen inside an electron-transparent “core” (Figure 2A,C,E). It should be noted that a nucleoid and its components were not observed in many *S. aureus* cells on ultrathin sections. In published works, the structure of *S. aureus* nucleoid was not described in detail and authors limited themselves to fixing its presence, and, judging by published photographs, nucleoid is poorly distinguishable in many bacteria strains [29,32,33]. Probably, nucleoid visualization depends on the cell growth phase; actively replicating DNA of cells in the logarithmic growth phase can be visualized better, than in cells in the stationary phase, which we studied in this work.

Examination of *S. aureus* cells after 0 and 4 h of incubation in saline did not reveal noticeable differences in their ultrastructure. We described ultrastructure of *S. aureus* cells incubated in saline in this section, in subsequent descriptions these cells will be designated as “normal” or “control”.

### 3.3. Ultrastructure of S. aureus Treated with Cationic Peptides

As expected, R9F2 or (KFF)3K peptides differently damaged *S. aureus* cells starting from a very early time—one minute after mixing of cell suspension with the peptides.

Incubation of *S. aureus* with R9F2 or (KFF)3K peptides caused visible ultrastructural changes in the intermediate layer and the cell membrane within 1 min (Figure 3), while cells in a whole looked as “normal”. The intermediate layer swelled and decreased in electron density, and the periplasmic space widened up to 8–9 nm (6–7 nm in control cells) when incubated with R9F2 peptide, these changes were seen along whole perimeter of the cells (Figure 3B). In contrast, (KFF)3K peptide caused significant thinning (up to 3–4 nm) of the intermediate layer, but only in areas adjacent to damaged cell membrane (Figure 3C,D).

Swelling of electron-transparent lipid layer was the main indicator of cell membrane damage with R9F2 and (KFF)3K peptides, since protein outer and inner layers are soldered with an intermediate layer and cytoplasm, and their changes are inconspicuous. Lipid layer developed areas with many undulations and swelled up to 6 nm (Figure 3B,D), while in control cells it was straight and had thickness not exceeding 4 nm (Figure 2B and Figure 3A). We cannot definitely say if the lipid layer swells, or if it separates (exfoliates) from the adjacent protein layers. In any case, a space widening up to 15 nm appeared between two protein layers of cell membrane under the influence of R9F2 and (KFF)3K peptides. A substance filling this space remains homogeneous and electron-transparent (Figure 3). Changes in the cell membrane lipid layer could be related to violation of the molecular bonds between layers, which could be induced by increasing frequency of flip-flop events, which have been noted under bacteria incubation with some AMPs [34].

Thus, R9F2 and (KFF)3K peptides showed an ability to interact with *S. aureus* cells within 1 min and to cause visible changes in ultrastructure of cell membrane and intermediate layer, while cytoplasm remained visually unchanged. The ultrastructural changes described above were not detected in all cells after 1 min of incubation with the peptides, unaltered cells were present in both suspensions, however in the case of treatment with R9F2 unaltered cells were scarce.

Cell wall of *S. aureus* incubated with R9F2 or (KFF)3K peptides did not show noticeable changes in electron density and homogeneity within the whole period of incubation (up to 4 h).

Subsequent incubation of *S. aureus* with peptides R9F2 or (KFF)3K for 15 min enhanced the changes in ultrastructure of bacteria cells. R9F2 peptides caused dramatic deformation of these cells, which lost their spherical shape and became irregular, with large outgrowths (Figure 4D,E). The cell wall thinned to 18–20 nm (~25 nm in intact cells), and this could be involved in a decrease in cell wall stiffness and, in turn, in the deformation of the cells. Such changes in the cell wall of *S. aureus* differ from the noted thickening of cell wall reported for various antibacterial agents [23,35,36]. The intermediate layer remained somewhat swollen after incubation for 15 min with R9F2 peptide and decreased in electron density like those observed 1 min after the incubation (Figure 3B and Figure 4I). The expansion of cell membrane lipid layer (Figure 4F) and its undulation were present in many bacterial cells at this time-point; however, not in all cells and more than a half of the population were devoid of these changes.

Incubation with the R9F2 peptide for 15 min led to significant and similar damage in most cells of *S. aureus*: it became structureless and “lumpy” and contained large “empty” electron-transparent regions and amorphous clumps of different electron density; the ribosomes were unrecognizable (Figure 4D,F and Figure 5B). The size of bacteria cells did not increase. Similar ultrastructural changes in cytoplasm were observed in the cells of *S. aureus* treated with different antibiotics and plant compounds [33].

When studying *S. aureus* treated with peptides in TEM, the difference in the number of affected cells and the degree of their damage is striking: the R9F2 peptide affects all cells, while the (KFF)3K peptide does not, this is clearly seen in Figure 4. The population of *S. aureus* incubated with the (KFF)3K peptide during the experiment was heterogeneous and contained cells with a “normal” structure, while the R9F2 peptide induced deformation and other changes in all bacteria cells. We have excluded “normal” *S. aureus* cells from the description, all information given in the article concerns cells with visible abnormalities of ultrastructure.

The degree of *S. aureus* cell alteration varied significantly upon incubation with (KFF)3K peptide for 15 min (Figure 4G–I), and image 4 H represents an example of the most affected cell. Deformation of single cells was slight, that could reflect less damaged cell wall, compared with the R9F2 peptide effect. The cytoplasm lost clear ribosomes but did not become amorphous and “lumpy”; no large electron transparent “empty” regions were seen, as in the case of incubation with R9F2 peptide. *S. aureus* cells altered with (KFF)3K peptide had small electron-transparent areas in the cytoplasm, sometimes shaped like “rays” radiating from the center (Figure 4G–I and Appendix A).

Data obtained showed that the R9F2 peptide causes fast and significant destructive changes in the cytoplasm of all *S. aureus* cells within 15 min, while action of (KFF)3K peptide was significantly weaker for the same time-point. These observations correspond to data on changes in *S. aureus* viability (Figure 1).

Incubation of *S. aureus* suspension with R9F2 or (KFF)3K peptides for 30 min led to significant decrease in cell viability: by 3 orders of magnitude for the first, and by 1.5 orders of magnitude for the second (Figure 1). These data are consistent with TEM observations: on ultrathin sections of *S. aureus* treated with the R9F2 peptide, more destroyed cells were observed than in samples of *S. aureus* under (KFF)3K treatment.

It is pertinent to note that the amount of completely destroyed cells on ultrathin sections depends on processing for TEM: repeated washings remove destroyed cells which weigh much less than normal cells. Most of the severely damaged cells, which were observed after 15 min of incubation, were destroyed in the next 15 min and were removed from the samples. Consequently, on ultrathin sections, cells incubated with R9F2 or (KFF)3K peptides for 30 min looked less damaged than after 15 min of incubation. The same elimination of destroyed cells took place in cases of all subsequent examined samples (45 min, 1, 2 and 4 h) due to continuing destruction of *S. aureus* cells evidenced by the decrease in their viability (Figure 1).

Examination of *S. aureus* cells in TEM after 30 min of incubation with R9F2 peptides (Figure 5A,B) revealed great structural polymorphism due to differences in deformation degree, in a mass of electron dense amorphous clumps and areas of “empty” cytoplasm inside the cells. The cell wall looked “normal”, but its thickness varied from 14 to 20 nm and was generally less than that of “normal” cells (~25 nm). Most important ultrastructural finding in *S. aureus* cells treated for 30 min with an R9F2 peptide was the absence of alterations in cell membrane: no expansion of lipid layer or undulations were observed, and the cell membrane looked as in control cells (Figure 2B and Figure 5B). It is possible to suppose that the noted damages of cell membrane (Figure 3B,C and Figure 4D) are of decisive importance for the viability of *S. aureus* cells, and all cells with an expanded lipid layer of cell membranes were destroyed within 30 min of incubation and removed during TEM-processing. Cell membrane of *S. aureus* kept a “normal” structure in all the studied time-points of the experiment with R9K2 peptide.

Most cells in ultrathin sections of *S. aureus* incubated with R9F2 peptide for 30 min had a high electron density due to the presence of areas of coagulated structureless cytoplasm. Electron transparent areas were relatively small and often contained fine electron dense grains (Figure 5B). Cells of *S. aureus* incubated with the (KFF)3K peptide also showed large polymorphism of ultrastructure, related mostly to the location and size of the amorphous clumps, areas filled with fine grains and areas of structureless cytoplasm (Figure 5C,D). As in the case of the R9F2 peptide, the cell membrane of *S. aureus* cells had a “normal” structure (Figure 5B). The main difference between cells incubated with the (KFF)3K peptide and those in cells incubated with the R9F2 peptide for the same time was the absence of “empty” areas in the cytoplasm.

Parameters of ultrastructural changes in *S. aureus* cells incubated with R9F2 or (KFF)3K peptides for 45 and 60 min visually remained as observed after 30 min of incubation, except of more a frequent appearance of mesosomes (Figure 5D).

The second hour of *S. aureus* incubation with peptides R9F2 and (KFF)3K led to significant changes in the ultrastructure of bacteria, which were observed for up to 4 h of incubation. The bacteria cytoplasm contained multilayered membrane structures varying by form and size (Figure 6A,C). All these structures could be divided into two types: (1) membranes fitted tightly together and had a high electron density; (2) membranes had an average electron density, and the space between them was filled with a homogeneous substance of average electron density that resembled the surrounding cytoplasm. We have identified these structures as having derived from mesosomes.

Formation of mesosomes and their derivatives under various influences was reported in many studies, multilayered membrane structures of different size and complexity developed at different time of treatment and was considered as a sign of *S. aureus* cell membrane alteration [15,22,23,33,35,37]. In particular, Raj and coauthors [36], using a set of five antibiotics, observed the formation of mesosomes in *S. aureus* cells exceptionally in presence of antibiotics. However, all studies recorded only the presence of mesosomes or multilayer membrane structures; data on their changes over time have not been published. Nevertheless, analysis of the published and our obtained data suggested that mesosomes initially form as invaginations of the cell membrane and are able to proliferate, and then their membrane loses its “normal” structure and becomes electron-dense, possibly forming a kind of repository of damaged cell membrane.

Another structure we found looked like a bundle of tangled thin threads (1–2 nm) with a high electron density (Figure 6B–D). Sizes and appearance of these threads resembled DNA fibers of a high electron density, decorated with electron dense small grains, and we called these structures “decorated” fibers. It is interesting that both multilayered membrane structures and “decorated” fibers were not unique findings: first or second structures, or both, were observed in each cell of *S. aureus* incubated with R9F2 or (KFF)3K peptides for 4 h. We did not find any references or images of such structures in published works on the ultrastructure of Staphylococcus under various treatments. We noted similar “decoration” of DNA with silver nanoparticles (1–2 nm) in *S. aureus*, while DNA of *S. typhimurium* in the same experiment did not bind with nanoparticles [27]. In this work, we observed “decoration” with fine electron dense grains accompanied by high electron density of DNA, and we think that this observation reflects the damage to DNA visualized due to the increase of its osmiophilia.

Multimembrane structures and “decorated” fibers were most bright features of *S. aureus* ultrastructure and occupied a fairly large part of the cell and shadowed other changes of bacteria cells incubated with both peptides. Cytoplasm in *S. aureus* cells was represented by amorphous clumps and masses having middle electron density (Figure 6), varying in size and location. The “empty” areas varied in size and shape, providing a variety of cell images on ultrathin sections.

Thus, R9F2 peptides rapidly and deeply affected all *S. aureus* cells in suspension, while (KFF)3K peptides induced weaker changes, which did not involve all cells and developed slower. The appearance of multimembrane structures and “decorated” fibers evidence for development of alteration in cell membrane and DNA at molecular level induced by R9F2 and (KFF)3K peptides. The amount of destroyed cells increased in samples treated with both peptides during the incubation, which corresponded to data on viability testing. *S. aureus* suspension after 4 h of incubation with (KFF)3K peptide showed greater structural polymorphism than those treated with the R9F2 peptide.

### 3.4. Ultrastructure of S. aureus Treated with Chlorhexidine

Chlorhexidine, as both cationic peptides, caused visible changes in *S. aureus* cells within 1 min of incubation (Figure 7A,B). The cell wall (Figure 7A) became loosened and its outer edge looked serrated, the thickness decreased to 20–21 nm (~25 nm in control). The intermediate layer also looked loosened and less electron-dense than in control cells. The damage to cell membranes was detected as a loss of its lipid layer structure and an increase in its electron density.

Bright features of chlorhexidine action on *S. aureus* cells within 1 min were electron-dense clumps with blurred contours (Figure 7A,B). The clumps, with TEM magnification, looked like shapeless area of cytoplasm “stained” in dark color, no “specific” grains or fibers were detected. It could be compared with histological stains, which colored, for example, mucous granules. Apparently, chlorhexidine, which has entered *S. aureus* cell, interact with unstructured components of the cytoplasm, which are localized in the form of compartments differing in size and shape. This interaction could change chemical properties of cytoplasmic components and make them osmiophilic. Accordingly, cytoplasm compartments become electron-dense.

*S. aureus*, like other bacteria, can produce membrane vesicles (MVs) (20–400 nm in diameter), which could influence various biological processes. It is believed that formation of MVs in gram-positive bacteria is closely related to the alteration of the cell wall, in which “holes” are formed, through which vesicles bulge out. Current ideas about mechanisms of formation and functions of MVs are reviewed in [38,39]. We detected MVs (20–50 nm) in ultrathin sections of *S. aureus* cell suspension after 1 min of incubation with chlorhexidine (Figure 7C,F), while in the control culture and in the culture under treatment with R9F2 and (KFF)3K peptides, no such MVs in any time-point were found. The vesicles were filled with electron-dense grains or were “empty” (Figure 7C,F), and were observed in all samples of *S. aureus* incubated with chlorhexidine during the whole experiment (Figure 7D, Figure 8C and Figure 9D). Obviously, their presence reflected damage to cell walls clearly observed in *S. aureus* under chlorhexidine influence.

Incubation with chlorhexidine for 15 min led to increased damage to *S. aureus* cell ultrastructure. The electron density of the intermediate layer decreased even more, and in some areas it could not differ from the wall material (Figure 7F and Appendix A). The lipid layer of the cell membrane thickened to 7–8 nm (4–5 nm in the control). The outlines of electron-dense clumps became clearer and their shape became roundish. The clumps were observed in each bacteria cell and persisted up to 4 h of incubation; they tended to increase in size and were more often located near the cell membrane and the division septum (Figure 7D).

Chlorhexidine disrupted septa formation, and a set of ultrastructural changes indicative of this effect was observed starting from 15 min of incubation up to 4 h. The formation of multimembrane structures, which did not resemble mesosomes, was found. These structures consisted of thin membranes and were located in the areas of septa formation and on ultrathin sections, which looked like a loose accumulation of unordered membranes sometimes containing electron-dense material (Figure 8A–C). Based on a comparison of electron microscopic images, we can conclude that chlorhexidine-induced multimembrane structures are not identical to those formed under cationic peptide influence, although all are derived from the cell membrane. Ordinary “mesosomes” (Figure 7E) were exceptionally rare in *S. aureus* cells during chlorhexidine incubation.

A huge variety of images of septal disturbances was observed on ultrathin sections of *S. aureus* incubated with chlorhexidine for 15 min, due to the different stages of septa formation. It should be noted that this variety was retained during whole experiment. Images in Figure 9B–D demonstrate the early septal disruption in the presence of chlorhexidine.

The effect of chlorhexidine was clearly visible when comparing images of complete septa in control cells and treated cells (Figure 10). Instead of straight septa within clearly seen tightly fitted cell walls (Figure 10A), widened electron transparent space containing deformed cell walls separated from bacteria cells are seen (Figure 10B–E). The image of the affected cell walls depended on an ultrathin section plain, and when a section came perpendicularly, the cell wall was clearly visible (Figure 10B).

Analysis of obtained data allowed us to conclude that chlorhexidine destroys septa at different stages of their formation, and we see a set of cells at different stages of division rather than successive stages of damage. Obviously, chlorhexidine disrupted the cell wall and cell membrane in areas of septa formation, which seem to be more vulnerable to chlorhexidine, as it was shown, for example, for penicillin [40].

The layer formed by electron-dense grains, detected under a *S. aureus* cell membrane after 15 min of incubation with chlorhexidine (Figure 7F) within the incubation time became thicker (up to 11 nm) and more prominent (Figure 8C and Figure 9C,D).

In addition to the above-described changes in *S. aureus* cells, it should be noted that there were no visible changes in cytoplasm within 2 h of incubation (except of electron dense clumps presence). Later, ribosomes, in most cells, lost their distinct structure and acquired a “halo” of small electron density (Figure 8C and Figure 9C,D), which gave the cytoplasm a mesh appearance. In general, the cytoplasm maintained its electron density and the “lumpy” structure and amorphous clumps were absent.

## 4. Discussion

Effective killing of various microorganisms by AMPs determines the interest in these compounds, which have been extensively studied over last decade. Many studies are devoted to the examination of AMPs interaction with bacterial cells in order to understand its mechanisms, but they still remain “in the fog”, primarily in relation to gram-positive bacteria [23,41].

In this work we examined the interaction of gram-positive *S. aureus* with cationic peptides R9F2 and (KFF)3K, differing, among other parameters, by the net charge and amphiphilicity [18]. Examination of ultrathin sections in TEM is often used to illustrate the result of the damage to bacteria treated with one or another compound. However, researchers usually examine the cells at one time-point, and try to show the maximum effect of the test compound [21,23,28]. We take a different approach and examine the cell destructive change at successive time points to trace their dynamics. This approach makes it possible to determine the initial changes in microorganism ultrastructure, which, as a rule, afterwards are masked by late changes. Thus, we detected the clearly visible instant effect of both peptides on ultrastructure of *S. aureus* and, surprisingly, the effect of chlorhexidine, which was used as a positive control. The different character of ultrastructural changes after 1 min of incubation proved a specificity of reagents action. Fast effects were observed when R9F2 and (KFF)3K peptides attacked cells of *C. albicans* [19]. Both *C. albicans* and *S. aureus* showed damage of cell membrane as a first event in the interaction with R9F2 and (KFF)3K, and this evidence suggests that the cell wall did not protect microorganisms from peptide action, regardless of cell wall thickness and chemical composition.

Cell wall integrity and functionality is critical for bacteria existence, and many antibiotics damage this structure [14,31,40]. The cell wall of *S. aureus* incubated with R9F2 and (KFF)3K peptides did not showed changes in electron density and homogeneity, however measurements revealed the thinning of the *S. aureus* cell wall under R9F2 influence (up to 18–20 nm, in control ~25 nm), associated with pronounced cell deformation (Figure 4D,E). We were unable to find publications reporting a similar deformation.

Clorhexidine has been used in medical practice for more than 70 years; however, no publications reporting its effect on ultrastructure of *S. aureus* and other microorganisms were found. Meanwhile, this compound caused “peculiar damage” to cytoplasm, cell wall and septa formation in *S. aureus* cells, which are described in this paper. In *S. aureus* cells sampled after 1 min of incubation with chlorhexidine, we observed a thinning of the cell wall, which looked “gnawed” from the outside, and its structure was somewhat loosened (Figure 7). The degree of cell wall damage varied along the cell perimeter, probably due to adsorption of different amounts of chlorhexidine molecules. Areas of septa formation were heavily affected with chlorhexidine, which could be explained by the known fact that the cell wall in *S. aureus* is polymerized exclusively at the septum, making this a target for antibacterial treatment [42]. We tried to compare the ultrastructural damage of *S. aureus* by chlorhexidine with the effect of other antimicrobial drugs, but we were unable to find similar images or descriptions in published works. At the same time, data indicating selective damage to the septal formation area were presented [40,43]. Thus, we revealed a different pattern in damage to the *S. aureus* cell wall by the R9F2 and (KFF)3K peptides and chlorhexidine, the latter causing significantly more pronounced changes. The cell wall of *C. albicans* also were affected by chlorhexidine, and ultrastructural changes clearly differed from those caused by R9F2 and (KFF)3K peptides [19]. Thus, the changes in cell wall ultrastructure upon treatment with chlorhexidine were different in *S. aureus* and *C. albicans*, which is apparently due to the different chemical composition of the walls.

It is important to note that the size of *S. aureus* cells remained the same under influence of R9F2 and (KFF)3K peptides, while the cytoplasm underwent pronounced changes, which could be summarized as a loss of structure, amorphization and coagulation (Figure 4 and Figure 5). The bacterial cells also contained more or less large “empty” electron-transparent areas, and we are inclined to believe that these areas appear due to the rapid coagulation of cytoplasm, when the formed clots “stick together” near the plasmalemma, thereby “freeing” central areas of the cell. In contrast, treatment with chlorhexidine did not cause such dramatic changes in *S. aureus* cytoplasm: it differed from those in control cells only by the formation of a “halo” around the ribosomes.

Mesosomes originating from bacterial cell membrane and having a different shape are considered to be a sign of damage to the cell membrane [35,37]. Mesosomes in *S. aureus* cells were the most striking and common feature of action of R9F2 and (KFF)3K peptides, ultrathin sections of all cells showed the presence of these membranous structures and many of them had a high electron density. Similar osmiophilic multimembrane structures have been observed in *C. albicans* cells; however, their formation followed damage to the cytoplasmic membranes or cell membrane [19], whereas in *S*. *aureus* cells, mesosomes were formed first, and osmiophilia appeared later. We assume that an increase in osmiophilia of mesosomes membranes occurs due to their oxidative damage, which develops after mesosomes formation, while in *C. albicans* formation of multimembrane structures is a way of eliminating already damaged membranes. These differences are undoubtedly associated with different metabolisms in pro- and eukaryotic cells.

In chlorhexidine-treated cells of *S. aureus*, “typical” mesosomes were not formed; however, large multimembrane structures formed by thin membranes were observed in relation to septa formation (Figure 8 and Figure 9). We suppose that their formation is a consequence of disturbances in septa formation.

Our study of *S. aureus* cells on ultrathin sections using TEM revealed distinct differences in the effect of R9F2 and (KFF)3K peptides, and chlorhexidine on bacterial ultrastructure. The first target for both peptides was plasmalemma, and its disruption (Figure 8 and Figure 9) was responsible for the death of *S. aureus* cells within the first 30 min of incubation, which is evidenced by the disappearance of cells, showing the damage of cell membranes later during the incubation. It is known that most antibacterial AMPs are cationic, and many are amphipathic, and it is also believed that these properties provide their ability to interact with the cell membrane of bacteria by binding to both negatively and positively charged molecules [44,45,46,47]. Many studies suggest that AMPs interact with the phospholipid bilayer of the bacteria cell membrane, which leads to the formation of pores and leakage of cytoplasm, and then death of bacteria [48,49]. However, observation of gram-positive bacteria raised the question: do AMPs pass through the bacterial cell wall without any interaction with its components?

Our study revealed changes in thickness of *S. aureus* cell walls after 1 min incubation with R9F2 and (KFF)3K peptides, supposing the interaction of these peptides with cell wall components. These data are consistent with a recently published first study examining the interaction of AMPs and peptidoglycan, which showed the interaction of melittin and cecropin A with peptidoglycan, a major component of bacterial cell wall [50].

We believe that the interaction of the cell wall of gram-positive bacteria with AMP is of great importance in understanding the mechanisms of AMP action on bacteria and should be taken into account when developing models of AMP action. This is important not only in terms of theory of the action of AMP on bacteria, but also in terms of the development of antimicrobial agents based on AMP, which is extremely important for overcoming the problem of antibiotic resistance by bacteria.

## Figures and Tables

**Figure 1 microorganisms-08-01991-f001:**
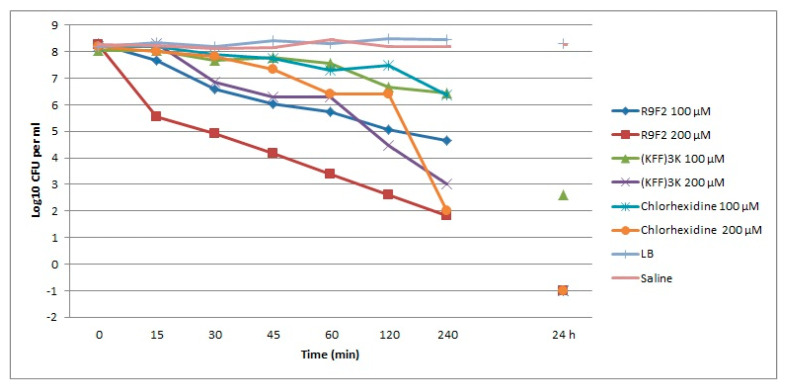
Viability of *S. aureus* in the presence of R9F2 and (KFF)3K peptides and chlorhexidine. Viability of *S. aureus* in 0.9% NaCl solution (saline) and LB broth is presented as a negative control.

**Figure 2 microorganisms-08-01991-f002:**
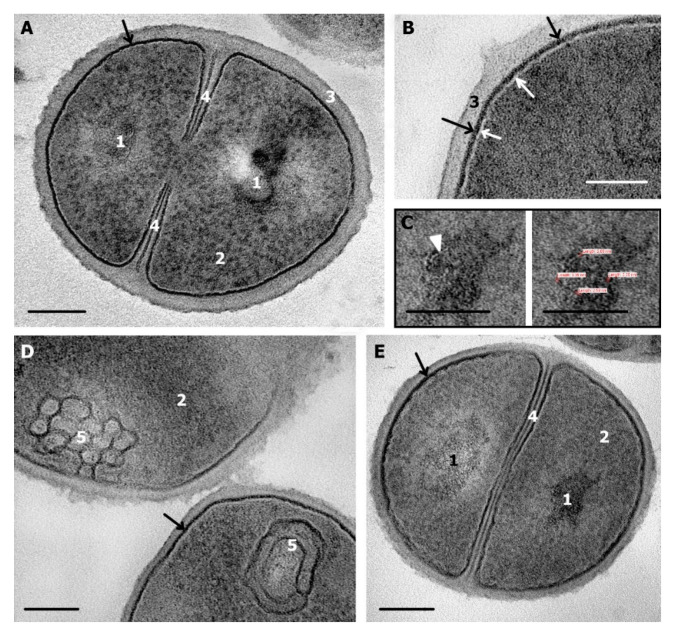
*S. aureus* control (normal) cells. (**A**) dividing cell; (**B**) structure of cell periphery; (**C**) DNA strands (thickness about 3 nm); (**D**) mesosomes in cytoplasm, note a contact with cell membrane in top cell; (**E**) cells after division. 1—nucleoid; 2—cytoplasm; 3—cell wall; 4—septum; 5—mesosome. Black arrows show the intermediate layer; white arrows show electron transparent lipid layer of cell membrane, white arrowhead—DNA. TEM of ultrathin sections. Scale bars correspond to: (**A**)—200 nm, (**B**–**E**)—100 nm.

**Figure 3 microorganisms-08-01991-f003:**
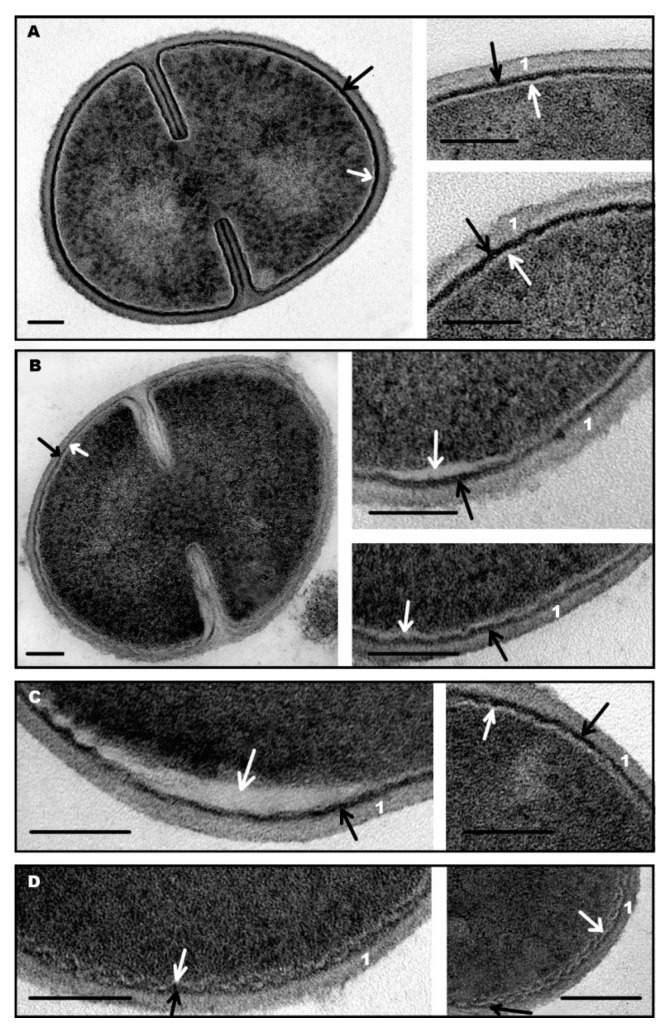
Ultrastructure of *S. aureus* control cells (**A**); cells after 1 min incubation with R9F2 (**B**) and (KFF)3K peptides (**C**,**D**). 1—cell wall. Black arrows show intermediate layer; white arrows—electron transparent lipid layer of cell membrane. TEM of ultrathin sections. Scale bars correspond to 100 nm.

**Figure 4 microorganisms-08-01991-f004:**
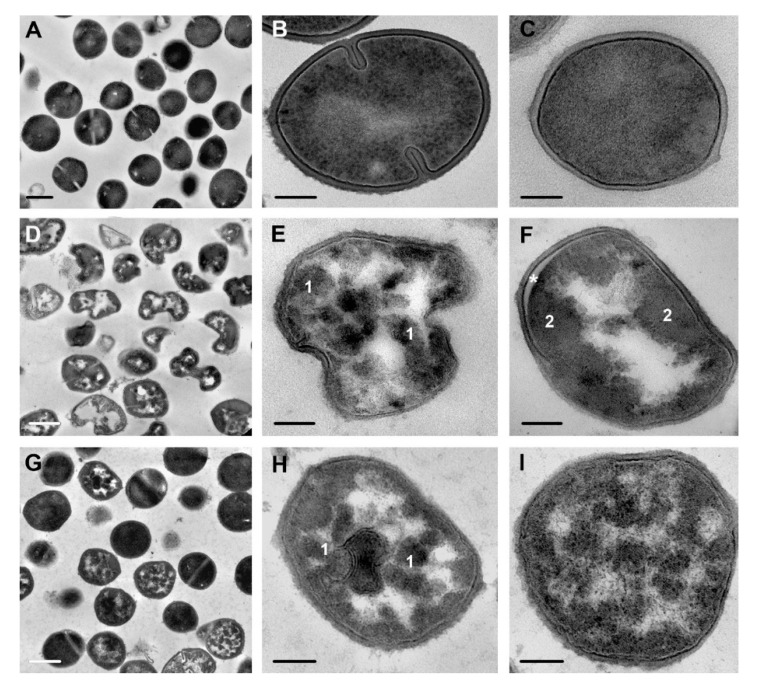
*S. aureus* cells: (**A**–**C**)—control cells; (**D**–**F**)—cells incubated with peptide R9F2 for 15 min, all cells are deformed and possess “empty” areas in cytoplasm. (**G**–**I**)—cells incubated with (KFF)3K peptide for 15 min, different degree of damage to cytoplasm is observed. Note the alteration of the intermediate layer and undulation of cell membrane lipid layer on image I. 1—amorphous clumps in cytoplasm; 2—electron dense amorphous cytoplasm at cell periphery; asterisk shows widening of lipid layer. TEM of ultrathin sections. Scale bars correspond to: (**A**,**D**,**G**): 500 nm, other 100 nm.

**Figure 5 microorganisms-08-01991-f005:**
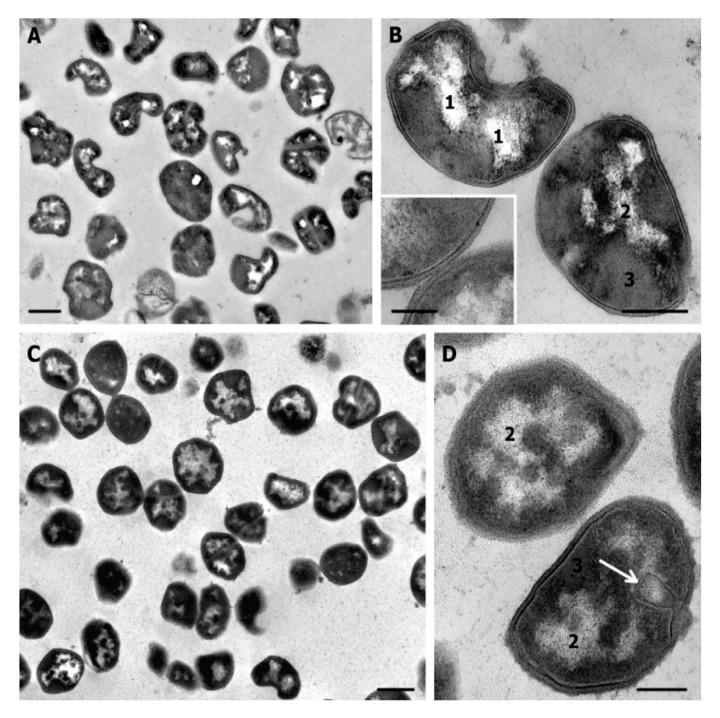
Ultrastructure of *S. aureus* cells incubated for 30 min with peptides R9F2 (**A**,**B**) and (KFF)3K (**C**,**D**). Insert: two cells, demonstrating “normal” cell walls. 1—“empty” areas, 2—areas filled with fine grains; 3—coagulated structureless cytoplasm. Arrow shows mesosome formation. TEM of ultrathin sections. Scale bars correspond to: (**A**,**C**)—500 nm, (**B**,**D**)—200 nm, inset—100 nm.

**Figure 6 microorganisms-08-01991-f006:**
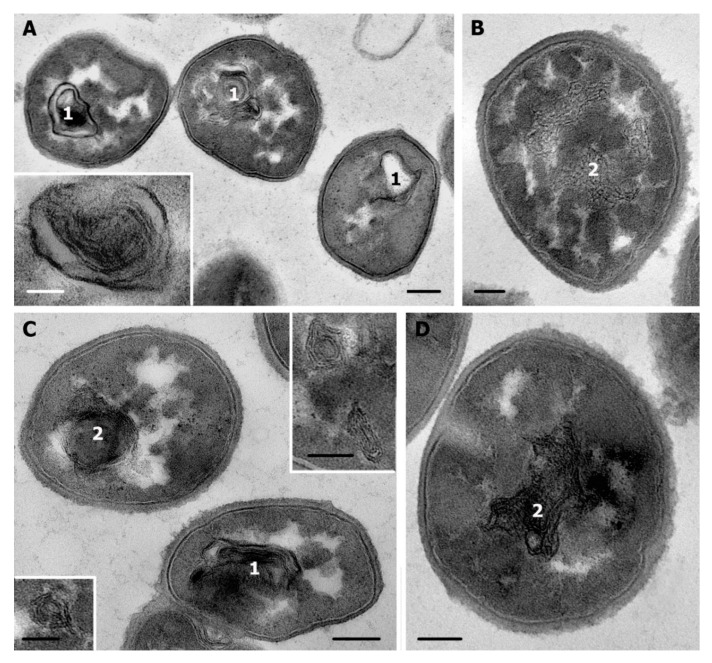
Cells of *S. aureus* incubated for 4 h with R9F2 (**A**,**C**) and (KFF)3K (**B**,**D**) peptides. 1—multimembrane structures, the insert on image A shows these structure at high magnification, 2—“decorated” fibers, inserts on image C shows “decorated” fibers at high magnification. TEM of ultrathin sections. Scale bars correspond to: (**A**,**C**)—200 nm, (**B**,**D**)—100 nm, insets—50 nm.

**Figure 7 microorganisms-08-01991-f007:**
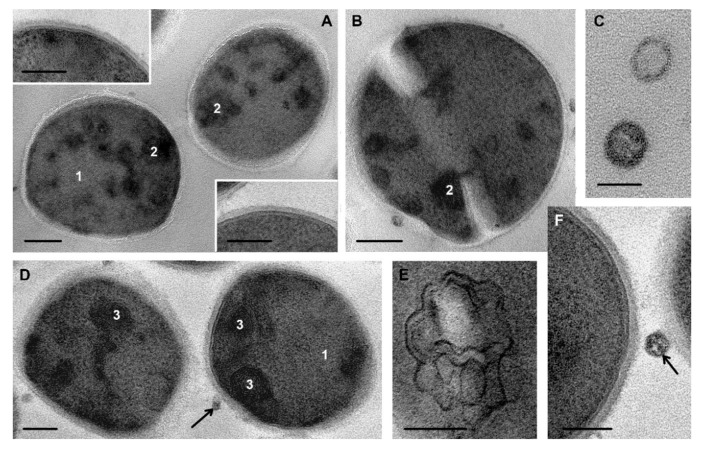
Cells of *S. aureus* incubated with chlorhexidine for 1 min (**A**,**B**) and 15 min (**D,F**). (**C**)—membrane vesicles, 1 min oh incubation. (**E**) mesosome. 1—cytoplasm, 2—electron dense clumps, 3—electron dense masses. Extracellular vesicles are shown with arrows. TEM of ultrathin sections. Scale bars correspond to: (**A**,**B**,**D**,**F**)—200 nm, (**C**,**E**)—100 nm, insets—100 nm.

**Figure 8 microorganisms-08-01991-f008:**
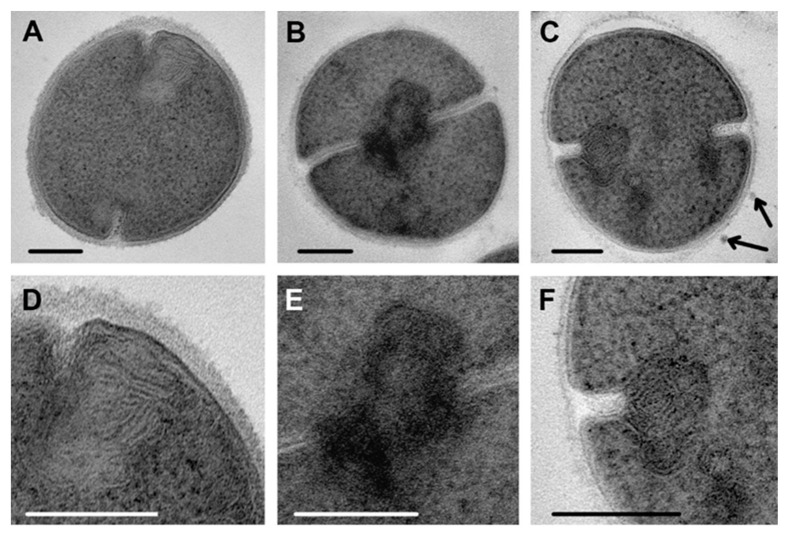
Multimembrane structures located in areas of septa formation in *S. aureus* cells incubated with chlorhexidine for 15 min (**A**); 1 h (**B**) and 4 h (**C**). (**D**–**F**)—enlarged fragments of images (**A**–**C**). Arrow shows MV. TEM of ultrathin sections. Scale bars correspond to 200 nm.

**Figure 9 microorganisms-08-01991-f009:**
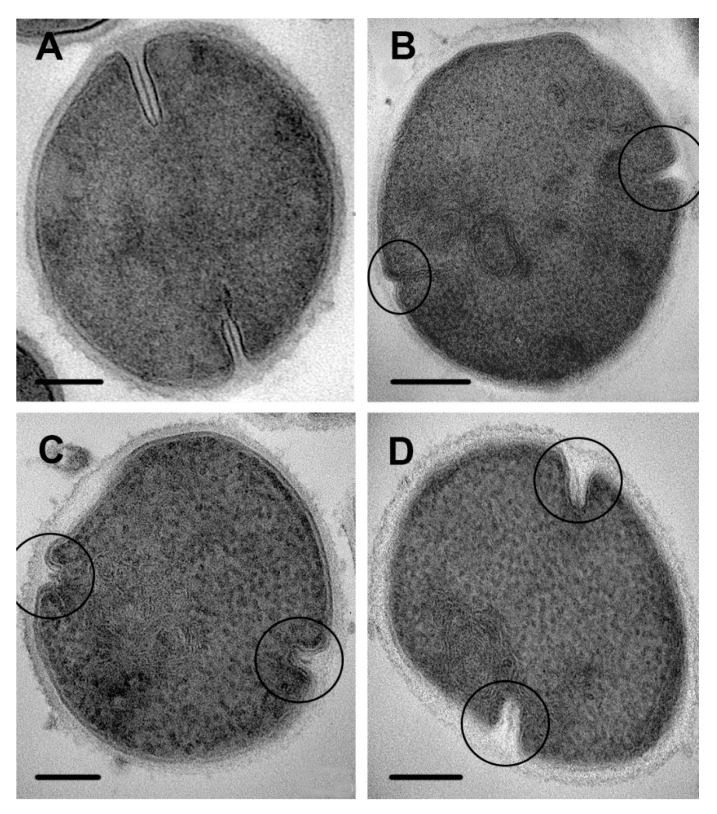
Cells of *S. aureus* incubated with chlorhexidine. (**A**) control cell; (**B**) 30 min incubation; (**C**,**D**) 120 min. Ovals show the areas of septum formation, arrows show the deposition of electron-dense material under cell membrane. TEM of ultrathin sections. Scale bars correspond to 200 nm.

**Figure 10 microorganisms-08-01991-f010:**
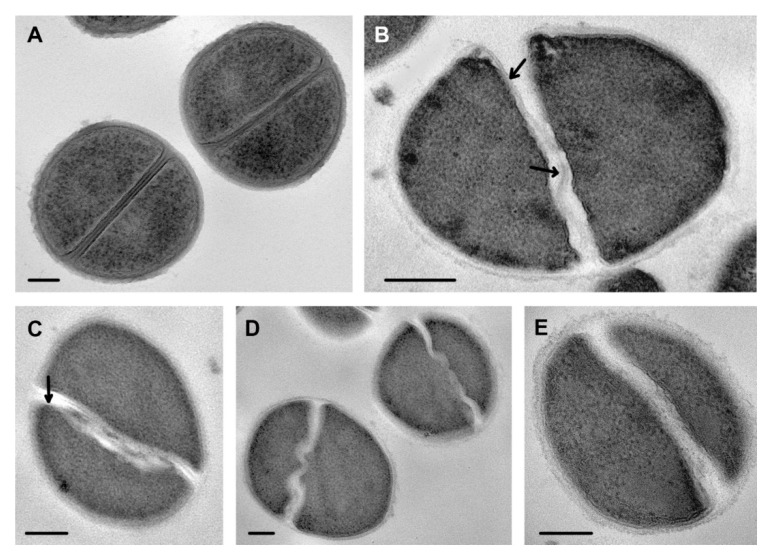
Damage to septa in *S. aureus* incubated with chlorhexidine. (**A**) control cells; (**B**) incubated for 15 min; (**C**,**D**) for 45 min; (**E**) 2 h. Arrows show cell wall in damaged septa. TEM of ultrathin sections. Scale bars correspond to 200 nm.

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
