# Peer review of "Changes in the Ultrastructure of Staphylococcus aureus Treated with Cationic Peptides and Chlorhexidine"

_microorganisms, 2020, doi:10.3390/microorganisms8121991_

Round 1

Reviewer 1 Report

The investigation of the antimicrobial peptides really interesting and important. So the described investigations are relevant. But the manuscript suffers from several mistakes, therefore some improvements are recommended.

In line 13 instead the two abbreviations the whole sequence should be given.

In line 40 The effect of AMP's are moreless known, but the mechanism many cases not.

In line 43 the practical use of AMP's do not depend from their origin.

In line 85 The two peptide structure is given an unusual way.

My main problems: As reference, to use of an antibiotic material would be more informative. Additionaly, some toxicity test of the investigated peptides to body cells would be important.

Author Response

Dear reviewer, we sincerely appreciate your interest to our work, and we are grateful that you took the time to work with our manuscript. Thank you for positive evaluation of our work.

Please find our responses to your comments.

  • My main problems: As reference, to use of an antibiotic material would be more informative. Additionally, some toxicity test of the investigated peptides to body cells would be important.

Thank you for these comments, we agree that comparison of peptides effect with antibiotics is interesting, however, we decided to postpone the comparison of R9F2 and (KFF)3K peptides with antibiotics to the next stage of our work, since the mechanism of peptide effect on cells was not clear. Besides, chlorhexidine was used as a reference in the study of R9F2 and (KFF)3K peptides effect on C. albicans.  

The main objective of this work was to study the ultrastructural changes in S. aureus cells under the influence of cationic peptides, and we relied on the results of a previously published study that showed no significant toxic and hemolytic effects of peptides on eukaryotic cell culture and mammalian erythrocytes [18]. Undoubtedly, deeper studies of the toxicity of peptides for cells at the level of the organism are needed, and we hope to get the opportunity to conduct them.

  1. Amirkhanov, N. V.; Tikunova, N. V. Synthetic Antimicrobial Peptides. I. Antimicrobial Activity of the Amphiphilic and Non-Amphiphilic Cationic Peptides Bioorganic chemistry 2018, 44, 5, 492-505.

In line 13 instead the two abbreviations the whole sequence should be given.

Corrected.

In line 40 The effect of AMP's are moreless known, but the mechanism many cases not.

line 40:

Effect of natural AMPs does not associate with bacteria specific molecules, and so development of bacterial resistance to AMPs is rare phenomenon, which increases the significance of these compounds [10].

New version:

Although many details of the action of natural peptides are still unknown, the collected data show that the action of natural AMPs is not associated with specific bacteria molecules, and therefore the development of bacterial resistance to AMPs may be a rare phenomenon, which increases the significance of these compounds.

In line 43 the practical use of AMP's do not depend from their origin.

Line 43:

The exceptional antimicrobial properties of natural AMPs are realized in living organisms, but their practical use in medicine is hampered by the difficulty of obtaining AMPs from natural sources, so synthetic AMPs are entering the arena as the "main players" [9,11].

This phrase is intended to explain why synthetic peptides appear in the foreground, and the difficulties in obtaining natural AMPs for medical use figure as one of the rationales.

We tried make the sentence more clear:

The exceptional antimicrobial properties of natural AMPs are realized in living organisms, but their practical use in medicine is hampered by the difficulty of obtaining AMPs from natural sources and standardization for medical application, so synthetic AMPs are entering the arena as the "main players" [9,11].

In line 85 The two peptide structure is given an unusual way.

Corrected.

New version:

The peptides H2N-(Lys-Phe-Phe)3-Lys-C(O)NH2 and H2N-Arg9-Phe2-C(O)NH2 were synthesized at >95% purity and verified by mass-spectrometry (Figure S1), in the text are designated as peptides (KFF)3K and R9F2.

Reviewer 2 Report

The manuscript presents the evaluation of different antimicrobial peptides on Staphylococcus aureus reporting terrific results on the effect of the two molecules (R9F2 and (KFF)3K) previously isolated (Amirkhanov et al., 2018). The authors evaluated their effect on the growth and through microscopy, the effect on the cell walls was assessed, demonstrating the action of the peptides. The experimental design is scientifically sound and the results are well presented, however, a few minor suggestions and corrections are needed for the final presentation of the data.

In particular:

Use consistently the italic for the name of bacteria (e.g. lines: 13, 15, 19, 21, 24, 30, 32, 62, 71 and so on) and keep consistent the format: C.albicans or C. albicans; S. aureus and St. aureus 

Lines 52-56: Please, the sentences are not completed, I believe the authors missed a few lines (is TEM the Transmission electron microscopy?)

please, check the format of the references and the standard for the journal (Lines 39 and 46) if the reference is a "review" and it is relevant for the section, explicit in the main text rather than beside the number of the reference.

Author Response

Dear Reviewer, we sincerely appreciate your interest to our work, and we are grateful that you took the time to work with our manuscript. Thank you very much for positive evaluation of our work.

Please find our responses to your comments.

  • Use consistently the italic for the name of bacteria (e.g. lines: 13, 15, 19, 21, 24, 30, 32, 62, 71 and so on) and keep consistent the format: albicansor C. albicans; S. aureus and St. aureus 

 We apologize for this negligence in the work on the manuscript, everything has been corrected.

Lines 52-56: Please, the sentences are not completed, I believe the authors missed a few lines (is TEM the Transmission electron microscopy?)

We added phrase:

The action of AMPs at population level usually is studied using microbiological and biochemical methods.

and inserted “full name” of the TEM:

The main tool for studying changes in cellular level is given by their submicron size, and this is transmission electron microscopy (TEM) of ultrathin sections, which allows you to analyze cell structures on a nanometer scale and gives an idea of what is happening inside the cell [13,14].

please, check the format of the references and the standard for the journal (Lines 39 and 46) if the reference is a "review" and it is relevant for the section, explicit in the main text rather than beside the number of the reference.

Thank you, it is corrected. 

Round 2

Reviewer 1 Report

The modifications by the authors mostly fulfils the requests, so I recommend the acception.

Author Response

Dear Reviewer, 

thank you very much for work with our manuscript! 

We appreciate your time and efforts. 

Sincerely yours, Elena Ryabchikova